# Rethinking the Uncertainty: A Critical Review and Analysis in the Era of Large Language Models

## Abstract

In recent years, Large Language Models (LLMs) have become fundamental to a broad spectrum of artificial intelligence applications. As the use of LLMs expands, precisely estimating the uncertainty in their predictions has become crucial. Current methods often struggle to accurately identify, measure, and address the true uncertainty, with many focusing primarily on estimating model confidence. This discrepancy is largely due to an incomplete understanding of where, when, and how uncertainties are injected into models. This paper introduces a comprehensive framework specifically designed to identify and understand the types and sources of uncertainty, aligned with the unique characteristics of LLMs. Our framework enhances the understanding of the diverse landscape of uncertainties by systematically categorizing and defining each type, establishing a solid foundation for developing targeted methods that can precisely quantify these uncertainties. We also provide a detailed introduction to key related concepts and examine the limitations of current methods in mission-critical and safety-sensitive applications. The paper concludes with a perspective on future directions aimed at enhancing the reliability and practical adoption of these methods in real-world scenarios.

## 1 Introduction

Large Language Models (LLMs) have recently demonstrated remarkable capabilities in various complex reasoning and question-answering tasks (Zhao et al., 2023; Wang et al., 2024c; Liang et al., 2022). However, despite their potential, LLMs still face significant challenges in generating erroneous answers (Ji et al., 2023a; Li et al., 2023a; Huang et al., 2023), which can have serious consequences, particularly in domains where high levels of accuracy and reliability are critical. A key issue undermining trust in LLM outputs is the models' lack of transparency and expressiveness in their decision-making processes (Zhou et al., 2023; Lin et al., 2023; Yin et al., 2023; Xiao & Wang, 2018; Hüllermeier & Waegeman, 2021), where comprehensively understanding and estimating the model's uncertainty plays a vital role. For example, in the medical field, a physician diagnosing a critical condition like cancer would not only require a high predictive accuracy from the model but also need to understand the uncertainty associated with the case (Gawlikowski et al., 2022a).

While the need for quantifying uncertainty in LLMs is widely recognized, there still lacks a consensus on the interpretation of uncertainty in this new context (Gawlikowski et al., 2022a; Mena et al., 2021; Guo et al., 2022; Hüllermeier & Waegeman, 2021; Malinin & Gales, 2018), which in turn further complicates its estimation. Terms such as "*uncertainty*", "*confidence*", and "*reliability*" are often used interchangeably, yet they refer to distinct concepts that require careful distinction (Gawlikowski et al., 2021). For instance, an LLM can exhibit a high-confidence response to an inherently uncertain and unanswerable question. However, this response could be contextually inappropriate or factually incorrect, illustrating that high confidence

does not necessarily correspond to low uncertainty (Gawlikowski et al., 2022b). Thus, the first challenge that remains in the literature is to *explicate the definition of uncertainty in the context of LLMs and explore the nuanced differences between these intertwined concepts*.

Traditionally, uncertainty in deep neural networks (DNNs) is categorized into two types: *aleatoric*, arising from data randomness such as sensor noise, and *epistemic*, stemming from limitations in model knowledge due to insufficient data or unmodeled complexities (Gawlikowski et al., 2022a; Mena et al., 2021; Guo et al., 2022; Hüllermeier & Waegeman, 2021; Malinin & Gales, 2018). Although these categories are widely used in deep learning, they do not fully address the unique challenges of LLMs, which include processing complex text data, managing extremely large parameters, and dealing with often inaccessible training data. Furthermore, the entire lifecycle of LLMs—from pre-training through inference—introduces unique uncertainties, as does the interaction between users and these models. Understanding these different sources of uncertainty is critical, particularly from the perspective of making LLMs more interpretable and robust. Achieving this understanding, however, is not possible without an *inclusive and fine-grained framework that systematically identifies and analyzes the various sources of uncertainty in LLMs*.

Recently, numerous studies have been proposed, aiming to estimate the uncertainty in LLMs (Manakul et al., 2023; Beigi et al., 2024; Azaria & Mitchell, 2023a; Kadavath et al., 2022; Kuhn et al., 2023), and can be broadly divided into four main categories based on their underlying mechanisms: logit-based (Lin et al., 2022b; Mielke et al., 2022a; Jiang et al., 2021; Kuhn et al., 2023), self-evaluation (Kadavath et al., 2022; Manakul et al., 2023; Lin et al., 2024a), consistency-based (Portillo Wightman et al., 2023; Wang et al., 2023), and internal-based (Beigi et al., 2024). However, given the unique characteristics and nuanced aspects of uncertainty in LLMs, critical questions arise regarding the effectiveness of each type of method in truly capturing uncertainly or other related aspects in the context of LLMs, and which specific sources of uncertainty are being detected across the various stages of an LLM's lifecycle. Answering these questions is vital for developing more reliable and comprehensive approaches to uncertainty estimation in LLMs.

To address the aforementioned challenges and questions, we conduct a critical review and analysis of studies related to uncertainty and other related concepts, aiming to present a comprehensive survey covering the full spectrum of uncertainty in LLMs, particularly focusing on the interplay between uncertainty concepts, sources, estimation methods, and text data characteristics, which, to the best of our knowledge, is still lacking in this field. In summary, our contributions in this survey are manifold and pioneering: (1) We have standardized definitions for uncertainty and explore related concepts, enhancing communication across the field (Section 2). (2) We are the first to propose a comprehensive framework that analyzes all sources of uncertainty throughout the lifecycle of LLMs, providing deep insights into their origins and effective management strategies (Section 3). (3) We evaluate and compare current methods for estimating and evaluating LLM uncertainty, discussing their strengths and limitations (Section 4). (4) Finally, we identify future research directions to enhance uncertainty estimation in LLMs, addressing critical gaps and emerging trends for improved reliability and accuracy in critical applications (Section 5).

## 2 DEFINITION OF UNCERTAINTY AND RELATED CONCEPTS

This section begins by offering a comprehensive definition of *uncertainty* and its associated concepts—*confidence* and *reliability*—within the context of large language models. As illustrated in Figure 1, although these concepts are interrelated, they pertain to distinct aspects of model performance that necessitate careful differentiation. We specifically emphasize the differences between *uncertainty* and *confidence*, terms that are often used interchangeably in the literature.

***Uncertainty*** fundamentally refers to the extent to which a model "**knows**" or "**does not know**" about a given input, based on the training it has received (Malinin & Gales, 2018; Der Kiureghian & Ditlevsen, 2009; Hüllermeier & Waegeman, 2021; Gawlikowski et al., 2022a; Kendall & Gal, 2017). Often, this arises

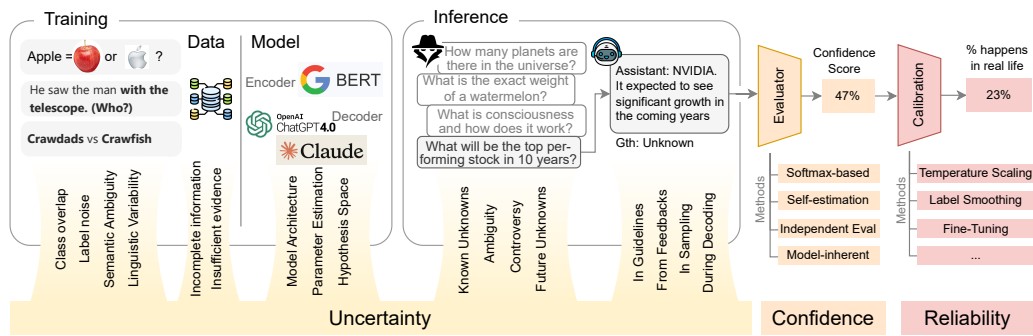

Figure 1: Visualization of the Distinct Aspects of Uncertainty, Confidence, and Reliability in Large Language Models

from inadequate or conflicting training data (Guo et al., 2022; Hüllermeier & Waegeman, 2021; Mena et al., 2021), inappropriate model selection (Gawlikowski et al., 2022a; Mena et al., 2021; Battaglia et al., 2018), or factors like noise and inherent data ambiguity (Kendall & Gal, 2017; Mena et al., 2021; Guo et al., 2022). These factors collectively delineate a model's understanding—or misunderstanding—of its operational environment, influencing the reliability of its outputs.

***Confidence***, often expressed as a "predicted probability score", quantifies the likelihood that a model's prediction is correct. Derived from the softmax output applied to logits, this score assigns each class a probability between 0 and 1, with the highest probability indicating the model's chosen prediction (He et al., 2024; Guo et al., 2017; Nandy et al., 2021). However, confidence scores can be misleading; they may exhibit "overconfidence", where the score is high despite inaccurate predictions, or "underconfidence", where scores are low even when predictions are correct (Lakshminarayanan et al., 2017; Wang, 2023; Chen et al., 2022; Guo et al., 2017).

***Reliability***. Merely estimating confidence scores is insufficient for safe decision-making. It's vital to align confidence scores with the actual probabilities of correct predictions, a process known as calibration (Guo et al., 2017; Wang, 2023). Re-calibration techniques have been developed to enhance this alignment, ensuring that confidence estimates are accurately calibrated and reliable for practical applications (Guo et al., 2017; Wang, 2023; Nixon et al., 2019; Mukhoti et al., 2020).

**Does High Confidence Score Always Mean Low Uncertainty?** Confidence scores are often interpreted as indicators of uncertainty, leading to significant challenges. DNNs frequently assign high confidence to inputs far removed from their training data, resulting in misleading confidence levels for incorrect classifications (Hein et al., 2019). For instance, a network trained on images of cats and dogs might confidently, but incorrectly, classify a bird as one of these categories (Gawlikowski et al., 2022a; Malinin & Gales, 2019). Therefore, high confidence does not necessarily indicate low uncertainty, making such an assumption problematic. LLMs also exhibit high confidence even when uncertainty is substantial. For example, models might confidently answer ambiguous or unanswerable questions like "*How many planets are in the universe?*" (**known unknowns**), "*What will be the top performing stock in 10 years?*" (**future unknowns**), "*What is consciousness and how does it work?*" (**controversial unknowns**), or "*What is the exact weight of a watermelon?*" (**ambiguous questions**), despite the inherent uncertainty of such questions. Similarly, models may express high certainty in speculative or hypothetical scenarios, like "*What would happen if the US had lost the Independence War?*", even when no definitive answer exists. These examples highlight that a high confidence score which derived through various computational methods, often imply as zero uncertainty, does not necessarily indicate the correctness of an answer. It is crucial, therefore, to approach high confidence scores with caution and to develop methods to measure the true uncertainty.

## 3 SOURCES OF UNCERTAINTY IN LARGE LANGUAGE MODELS

### 3.1 A COMPREHENSIVE FRAMEWORK FOR UNDERSTANDING UNCERTAINTY IN LLMS

There are three traditional categories of uncertainty commonly used in deep learning, including (1) **Model (epistemic) Uncertainty**, which pertains to uncertainties in estimating model parameters, reflecting model fit and its limitations in generalizing to unseen data (Der Kiureghian & Ditlevsen, 2009; Lahlou et al., 2023; Hüllermeier & Waegeman, 2021; Malinin & Gales, 2018); (2) **Data (or aleoteric) Uncertainty** that stems from complexities within the data itself, such as class overlap and various types of noise (Der Kiureghian & Ditlevsen, 2009; Rahaman & Thiery, 2020; Wang et al., 2019; Malinin & Gales, 2018); and (3) **Distributional Uncertainty**, which often dues to dataset shift and occurs when training and testing data distributions differ, leading to potential generalization issues during real-world applications where the model faces data markedly different from what it was trained on (Malinin & Gales, 2018; Nandy et al., 2021; Gawlikowski et al., 2022a; Chen et al., 2019; Mena et al., 2020).

These traditional uncertainty categories, though prevalent in deep learning, fail to fully address the unique challenges of LLMs. LLMs are characterized by extensive parameters, complex text data processing, and often limited access to training data, introducing specific uncertainties in their outputs. Moreover, interactions with users in dynamic environments and human biases in data annotation or model alignment complicate the uncertainty landscape. Unlike general deep learning models that primarily predict numerical outputs or classes, LLMs generate knowledge-based outputs which may include inconsistent or outdated information (Lin et al., 2024b). These features cannot be adequately addressed by simply categorizing uncertainty into three traditional types. These distinctive aspects necessitate a comprehensive framework to better understand the diverse sources of uncertainty in LLMs.

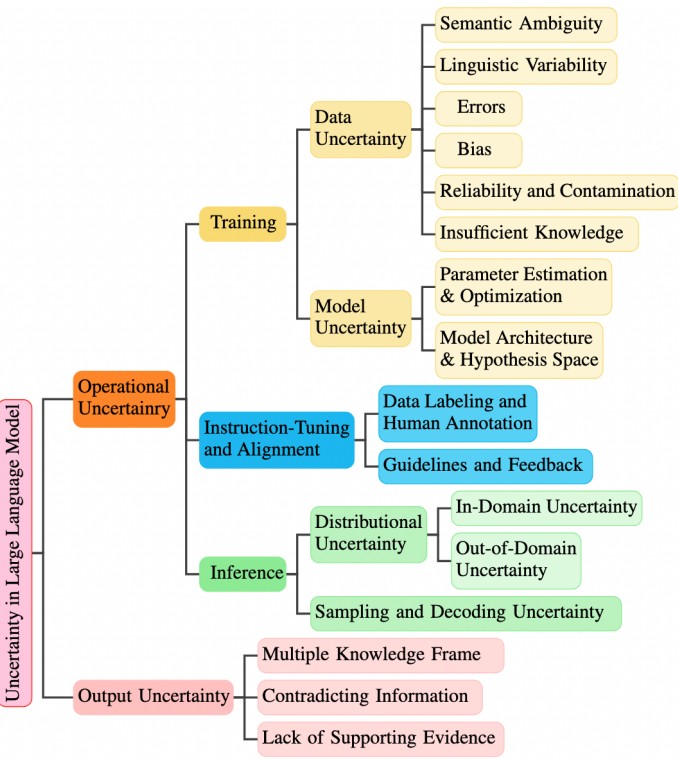

Figure 2: A comprehensive framework to categorize the sources of uncertainty through LLM's life-cycle.

To address these challenges, we introduce a new framework to categorize uncertainty in LLMs, as illustrated in Figure 2. This framework distinguishes between *operational* **uncertainty**, which pertains to model and data processing, and *output uncertainty*, focusing on the quality of the generated content. Specifically:

*Operational uncertainty* in LLMs arise from pre-training to inference, encompassing data acquisition, model and architecture design, training and optimization processes, alignment, and inference activities. These uncertainties stem from how LLMs are trained on extensive datasets, process inputs, and generate

text. In essence, operational uncertainties arise when the model is unable to capture the full complexity of the data it has been trained on, or when the input data itself introduces ambiguities or noise.

***Output uncertainty*** in LLMs stems from challenges in analyzing and interpreting the generated text, relating to the quality and reliability of information used for decision-making. For example, in medical scenarios, where an LLM is tasked with providing diagnostic suggestions based on patient symptoms, the model may generate multiple possible diagnoses. However, if these suggestions lack justification or supporting evidence, or contain contradictory information, significant uncertainty arises for the physician who must determine the credibility of these diagnoses. The physician may face substantial challenges in deciding which diagnosis to investigate further, highlighting the critical need for LLMs to provide well-supported, consistent, and reliable outputs to ensure their practical utility in decision-making processes.

By distinguishing between operational and output uncertainties, our framework offers several benefits: first, a fine-grained approach that captures the unique features of LLMs, providing a precise reflection of uncertainty for better modeling and understanding. Second, it establishes a foundation for identifying uncertainty sources, essential for developing targeted methods to accurately quantify them. Third, it offers stakeholder-specific insights, helping developers, users, and administrators address uncertainties relevant to their roles, enhancing model robustness, user interaction, and governance. Lastly, by aggregating beliefs and evaluating output evidence, the framework builds trust in LLM outputs, particularly in critical fields like medical diagnosis or legal reasoning.

## 3.2 OPERATIONAL UNCERTAINTY IN LLMS

### 3.2.1 OPERATIONAL UNCERTAINTY IN PRE-TRAINING AND TRAINING STAGE OF LLMS

**Data Uncertainty**    This phase involves collecting and organizing the pre-training corpus, which is critical as the quality, diversity, and representativeness of the data directly affect the model's understanding and ability to generalize. The sources of uncertainty at this stage include:

*(1) Semantic Ambiguity:* Textual data inherently contain semantic complexities, leading to significant uncertainties in training and inference processes for language models. For example, the word 'bank' can mean a financial institution or a river's edge, and 'lead' can refer to the act of leading or the metal, depending on the context (Anand & Kumar, 2022; Ott et al., 2018; Dreyer & Marcu, 2012; Blodgett et al., 2020). These semantic ambiguities pose challenges in maintaining meaning across different contexts and highlight the difficulty of achieving consistent semantic understanding. Such ambiguities are a primary source of uncertainty, complicating the model's understanding (Anand & Kumar, 2022; Piantadosi et al., 2011).

*(2) Linguistic Variability:* Textual environments are dynamic and subject to contextual and cultural shifts that significantly alter data interpretation and relevance (Kutuzov et al., 2018; Levy, 2008). For example, word meanings and usages evolve, new slang emerges, and topics range from casual conversations to specialized discussions, each with distinct linguistic nuances (Levy, 2008; Liu et al., 2018; Kutuzov et al., 2018). This variability requires language models to continually interpret context to determine meaning, greatly increasing the uncertainty in their knowledge and response.

*(3) Errors:* The data collection process can introduce errors like typographical mistakes, incorrect tagging, or grammatical errors. These inconsistencies can significantly mislead the learning process, impairing the LLM's ability to model and generate text accurately (Wang et al., 2024a; Liu et al., 2018).

*(4) Insufficient Coverage:* This refers to situations where incomplete coverage in the training dataset leads to uncertainty. Mitigating this requires acquiring more extensive and diverse data that encompasses various viewpoints (Gawlikowski et al., 2022a).

*(5) Reliability and Contamination of Data:* Training data for LLMs often contains inaccuracies or outdated content (Lin et al., 2024b), which can lead these models to perpetuate and amplify such errors (Lin et al.,

2024b). Misrepresented facts and contaminated data—incorrect or misleading information included in the training set—introduce significant uncertainty and hinder the models' reliability as decision-making tools (Jiang et al., 2024).

*(6) Human Biases:* In training data related to gender, race, socio-economic status, age, or disability are primary sources of uncertainty in LLM predictions (Bender & Friedman, 2018). These biases skew the model's understanding and responses, resulting in outputs that may not be universally valid or appropriate, thus increasing uncertainty about the model's performance and reliability in diverse real-world scenarios (Kirk et al., 2021).

**Model Uncertainty**   This type primarily arises from the model's fit to the data, highlighting its ability to generalize from the training data to unseen data (Malinin & Gales, 2018). The design of the architecture and the training process of an LLM are crucial in shaping its capabilities and effectiveness. This process involves the strategic configuration of the neural network, where each decision reflects an inductive bias—the underlying assumptions embedded in the model through choices in network structure. These biases influence how the model interprets and processes information (Battaglia et al., 2018). The sources of uncertainty pertain to model uncertainty are:

*(1) Model Architecture and Hypothesis Space:* The architecture of an LLM and the hypothesis space it explores are crucial to its performance and error susceptibility. Architectural decisions, such as the number of layers and network types, significantly influence model effectiveness across various tasks. These choices determine the hypothesis space—what the model can learn and predict—thereby introducing uncertainty in the model's ability to understand and generate language under different conditions. Variability in architectural setup can cause performance discrepancies when applied to new or varied datasets (Fedus et al., 2022; Abdar et al., 2021b; He et al., 2024; Gawlikowski et al., 2022a; Dodge et al., 2020).

*(2) Parameter Estimation and Optimization:* Parameter estimation and optimization methods are critical sources of variability and uncertainty in LLMs. Choices in optimization techniques (e.g., SGD, Adam), learning rates, loss functions, and regularization methods (e.g., dropout, L2 regularization) significantly impact the model's generalization capabilities and robustness (Lakshminarayanan et al., 2017). These factors contribute to uncertainty in the model's ability to consistently replicate results across different runs or datasets, affecting its adaptability to new data and stability across various operational environments (Payzan-LeNestour & Bossaerts, 2011).

### 3.2.2   OPERATIONAL UNCERTAINTY IN INSTRUCTION TUNING AND ALIGNMENT STAGE OF LLMS

Instruction-tuning and Reinforcement Learning from Human Feedback (RLHF) are advanced techniques that enhance LLMs' adaptability and responsiveness to specific tasks or user preferences (Ouyang et al., 2022; Rafailov et al., 2024). These techniques refine model responses to align more closely with expected outcomes using predefined instructions or guidelines (Bai et al., 2022; Askell et al., 2021). This process introduces uncertainty through two main sources:

*(1) Inconsistency and Bias in Data Labeling and Human Annotation:* Human labeling and annotation are fundamental sources of uncertainty in the training LLMs (Ghandeharioun et al., 2019; Abdar et al., 2021a; Zhou et al., 2024). The subjective nature of human judgment introduces variability and biases, affecting learning outcomes from reward models (Zhang et al., 2023a). Individual differences in perception and decision-making can lead to cognitive biases and inconsistency in labeled data, which RLHF reward modeling algorithms may exacerbate (Wang et al., 2024b; Denison et al., 2024).

*(2) Interpretation of Guidelines and Feedback:* The interpretation of instructions can vary, depending on the clarity of the guidelines and the model's ability to interpret them contextually Wang et al. (2024a). Discrepancies in understanding or applying these instructions can lead to variability in the model's outputs (Siththaranjan et al., 2023; Wu et al., 2024; Chidambaram et al., 2024; Park et al., 2024). The subjective

nature of feedback and its interpretation by the model can introduce additional layers of uncertainty. This is particularly evident when there is a lack of consensus among human reviewers, leading to challenges in achieving stable and predictable model behavior (Ghandeharioun et al., 2019; Abdar et al., 2021a; Zhou et al., 2024).

### 3.2.3 OPERATIONAL UNCERTAINTY IN INFERENCE STAGE OF LLMS

**Distributional Uncertainty** occurs when there are discrepancies between the training data distributions and those encountered during testing, a phenomenon known as dataset shift. This uncertainty is prevalent in real-world applications where models face data significantly different from their training sets. Distributional uncertainty indicates a lack of model familiarity with new data, posing challenges in making accurate predictions. This uncertainty is categorized into in-domain and out-of-domain types.

*(1) In-Domain Uncertainty:* This type of uncertainty occurs when LLMs operate within their training environments and inputs closely resemble the training data distribution. Such scenarios often present interpolation challenges or deal with 'unknown knowns' (Ashukha et al., 2021; Hüllermeier & Waegeman, 2021). Despite the similarity to training datasets, subtle nuances and variations within seemingly familiar data can still provoke uncertainties if not fully captured during training (Kim et al., 2023; Kong et al., 2020).

*(2) Out-of-Domain Uncertainty:* This occurs when LLMs face queries or data points outside their training distribution, leading to 'unknown knowns' and 'unknown unknowns,' where the model lacks the necessary data or precedent to generate well-founded responses, often resulting in overly generic or shallow outputs (Xu & Ding, 2024; Liu et al., 2024; Kong et al., 2020). 'Unknown knowns' are situations where the model has indirect knowledge but encounters data that, while potentially interpolatable from known data, still lies outside its direct experience, leading to uncertain responses despite possible correctness (Amayuelas et al., 2023). Conversely, 'unknown unknowns' refer to entirely unfamiliar data types or topics that the model has never encountered, typically producing speculative, erroneous, or hallucination.

**Sampling and Decoding Strategy** is another importatn sources of uncertainty in inference stage originate from the . The configuration of LLMs at inference time, including temperature scaling, context length (Anil et al., 2022), and decoding strategies such as beam search or nucleus sampling, significantly affects the uncertainty of model outputs. Temperature scaling adjusts the randomness of predictions by modifying the probability distribution, with lower values resulting in more deterministic outputs and higher values increasing diversity and variability. The choice of context length can influence the extent of uncertainty in outputs, with longer generations potentially introducing more uncertainty than shorter ones. Decoding strategies like beam search enhance output coherence by considering multiple possibilities, yet may reduce variability and creativity. These configurations are crucial for the model's generalization across tasks, impacting the coherence and consistency of predictions, thus playing a key role in balancing performance with uncertainty (Renze & Guven, 2024; Xie et al., 2023; Zeng et al., 2021; Ott et al., 2018; Hashimoto et al., 2024; Stahlberg & Byrne, 2019; Eikema & Aziz, 2020; Meister et al., 2020; Fan et al., 2018; Holtzman et al., 2020; Hewitt et al., 2022).

### 3.3 OUTPUT UNCERTAINTY IN LLMS

In contrast to operational uncertainties, which stem from the mechanics of how LLMs as deep neural networks generate text, output uncertainties focus on the outputs these models produce when used as knowledge generation tools. This capability can release a vast flow of information useful for decision-making in various downstream tasks. Here, the challenge shifts from a shortage of information to the risks of poorly understanding and managing inherent uncertainties, which may arise from unreliable, incomplete, deceptive, or conflicting information. The topic of reasoning and decision-making under uncertainty has been extensively explored in various AI domains, such as belief/evidence theory and game theory. This extensive knowledge base is crucial for enhancing our understanding of LLM outputs and identifying their inherent uncertainties,

important for tasks relying on this knowledge. Building on insights from a recent survey on uncertainty and belief theory (Guo et al., 2022), we adapted their framework to classify uncertainties better suited to the characteristics of LLM outputs. As a result, we categorize the sources and causes of output uncertainties based on the ***ambiguity*** in the output itself, which can stem from various causes and sources as:

***(1) Lack of supporting evidences and incomplete knowledge:*** An LLM may produce outputs with uncertainties due to a lack of supporting evidence in the responses, a common occurrence even in well-trained models addressing complex or nuanced topics. This uncertainty stems from the model generating conclusions without sufficient evidence to substantiate its answers, and from its inability to provide sufficient theoretical understanding or reliable information. The connections between claims and supporting information may not be clearly established or detailed, which hampers confident and reliable reasoning and decision-making. To mitigate this, the model's output can be enriched by incorporating more robust evidence or discarding unreliable evidence. Additionally, the complexity of model-generated information can overwhelm users due to limited cognitive capacity to process dense or intricate data. Simplifying the data into more manageable chunks with coarser granularity or focusing on key features while omitting less critical details can help. Effectively managing this uncertainty involves concentrating on leveraging the most relevant information available, thus enhancing the confidence and trust in the accuracy and relevance of the outputs.

***(2) Multiple knowledge frames and contradicting knowledge :*** These sources of uncertainty arise in scenarios where the same information—such as evidence or opinions—can be interpreted in various ways, leading to conflicting views. Multiple, valid beliefs about certain knowledge or information may coexist, often due to conflicting evidence. Conflicts can occur when parts of the information are incorrect, irrelevant, or when the model interpreting the data is not suitable for the current context. Additionally, conflicts may arise where there is no definitive ground truth, or in cases of controversial debate. Differing opinions from users, based on subjective perspectives, further complicate understanding and increase the layers of uncertainty.

## 4 APPROACHES FOR ESTIMATING AND EVALUATING UNCERTAINTY IN LLMS

Existing approaches to assess how well a model understands and is certain about its predictions in LLMs can be summarized into four major categories:

**Logit-based** approaches (Lin et al., 2022b; Mielke et al., 2022a; Jiang et al., 2021; Kuhn et al., 2023) assess model confidence by analyzing the probability distributions or entropy of outputs, providing clear measures of confidence. Although straightforward to implement, a fundamental issue with using logits as confidence indicators is that they reflect the probability distribution across potential tokens (vocabulary space), capturing linguistic forms rather than verifying the truthfulness or correctness of statements (Lin et al., 2022b; Si et al., 2022; Tian et al., 2023). Logit probabilities, irrespective of their magnitude, predominantly represent distribution over vocabulary space. Therefore, logit probabilities do not directly indicate model uncertainty but also reveal various linguistic factors that influence the model's output. This nuanced perspective is in stark contrast to human expressions of certainty, which typically reflect a belief in the accuracy or truth of a claim, based on information processing and decision-making processes, and are not influenced by phrasing (Koriat et al., 1980; Fischhoff et al., 1977). Additionally, another significant limitation is that logit-based methods do not identify or measure any types of uncertainty, limiting their applicability in scenarios where understanding the sources and degrees of uncertainty is crucial.

**Consistency-based** approaches (Vazhentsev et al., 2023; Portillo Wightman et al., 2023; Wang et al., 2023; Shi et al., 2022; Manakul et al., 2023; Agrawal et al., 2023) assess confidence by evaluating the agreement among various model responses, identifying potential inconsistencies. However, these methods encounter significant challenges, especially due to the diversity of potential paraphrases and formatting variations in textual data, complicating their use in real-time scenarios (Xiong et al., 2024; Jiang et al., 2021; Fadaee et al., 2017; Li et al., 2022; Ding et al., 2024; Kuhn et al., 2023). Additionally, a non-trivial challenge is the

effective measurement of consistency among responses, a persisting issue that hinders accurate confidence assessment (Manakul et al., 2023; Zhang et al., 2023b).

**Self-evaluation** methods (Kadavath et al., 2022; Manakul et al., 2023; Lin et al., 2024a) enable models to internally assess the correctness of their answers by leveraging their introspective capabilities. These methods employ various prompts that encourage models to express their confidence through numerical values or verbalized terms. Recent studies have refined these approaches, utilizing strategies like Chain of Thought (CoT) to enhance how models calibrate and articulate linguistic confidence (Xiong et al., 2023). Research has also explored expressing confidence with linguistics qualifiers to better align verbal expressions with the model's actual confidence levels (Mielke et al., 2022b; Zhou et al., 2023; Lin et al., 2022a), making model outputs more understandable for users. Despite their potential, these methods are constrained by the model's limited self-awareness, which can lead to circular reasoning and overconfident inaccuracies (Ji et al., 2023b; Chen et al., 2023). Another challenge is the interpretability and validity of obtained probabilities that align with specific linguistic and psychological interpretations, including expressions like 'I think,' 'undoubtedly,' or 'high confidence.'

**Internal-Based Approach** Recently, Beigi et al. (2024) used a mutual information framework to theoretically demonstrate that the internal states of large language models provide additional insights into the correctness of their answers. Burns et al. (2023) introduced an innovative unsupervised method that maps hidden states to probabilities. This approach involves responding to "Yes" or "No" questions, extracting model activations, converting these activations into probabilities. Furthering this research, studies have employed linear probes (Li et al., 2023b; Azaria & Mitchell, 2023b) and contrastive learning (Beigi et al., 2024) to assess whether the internal states across various layers can distinguish between correct and incorrect answers. Empirical results suggest that certain middle layers and specific attention heads show strong discriminative abilities. Beigi et al. (2024) expanded these findings by illustrating that for tasks requiring contextual processing, such as reading comprehension, the outputs of multi-head self-attention (MHSA) components are crucial for assessing response correctness. However, current methodologies exhibit limitations. Each task and dataset requires training a specific "confidence estimator" model, which restricts their generalizability. This limitation is evident as the performance of these methods often declines when trained on one task and dataset and tested on another, highlighting their limited transferability across different applications (Bashkansky et al., 2023). Additionally, the computational resources required to train these confidence estimators pose challenges for their deployment in real-time applications, further complicating their practical utility.

**What are the main properties of these methods in estimating uncertainty in LLMs?** Table 1 outlines the key characteristics of the methods discussed in this study, including their complexity, computational effort, memory consumption, flexibility, and their ability to identify and measure sources of uncertainty in LLMs.

| Description | Logit-Based | Internal-Based | Self-Evaluation | Consistency-Based |
|---|---|---|---|---|
| Uncertainty/Confidence | Confidence Score | Confidence Score | Confidence Score | Confidence Score |
| Identifying Sources | No | No | No | No |
| Need Access to Parameters | Yes | Yes | No | No |
| Explainability | No | somehow | No | No |
| Transferability | High | Low | High | Mid |
| Evaluation Metrics | Acc, ECE | Acc, ECE | Acc, ECE | Acc, ECE |
| Accuracy | Low | High | Very Low | Low |
| Need Training? | No | Yes | some methods | No |
| Comp. Effort Training | Low | High | Mid | Mid |
| Mem. Consumption Training | Low | High | Low | Mid |
| Comp. Effort Inference | Low | High | Low | Mid |
| Mem. Consumption Inference | Low | High | Low | Low |

Table 1: An overview of the four general methods presented in this paper. The labels *high* and *low* are given relative to the other approaches and based on the general idea behind them.

## 5 DISCUSSION AND FUTURE DIRECTION

**Go beyond Confidence Estimation:** As discussed, the literature on uncertainty estimation in LLMs primarily interprets confidence scores as measures of uncertainty. Such a prevalent method oversimplifies the nuanced and complex nature of uncertainty inherent in model predictions, which is crucial for accurate interpretation and reliability of model outputs. The limitations inherent in current methodologies necessitate the development of a more advanced framework for categorizing uncertainty estimation in large language models that surpasses the reliance on simple confidence scores.

**Lack of Explainability:** Current methods of confidence estimation provide certainty predictions without elucidating the underlying causes of potential uncertainties. While these scores may seem reasonable to human observers, the absence of insight into the sources of uncertainty complicates trust in the model's outputs, particularly in safety-critical contexts where explainability is essential. Current confidence quantification techniques struggle to pinpoint specific weaknesses or improvement areas in the model. Additionally, these methods lack the necessary transparency to clarify the reasons for model uncertainty, whether due to input ambiguity from users, insufficient knowledge, or conflicting information in the training data.

**Lack of Ground Truth for Uncertainty Estimation:** Current methods for estimating uncertainty are empirically evaluated and assess how accurately they predict the correctness of an answer. However, there is generally no established ground truth for validation, particularly regarding the sources of uncertainty, meaning currently there is no metric and method to determine the contribution of different uncertainties for specific models and tasks.

**Lack of Transferability of Uncertainty Estimation Methods Across Different Applications and Datasets:** Current uncertainty estimation methods often struggle with adaptability and generalizability when applied to new applications or datasets. Although effective within their specific domains, these methods frequently fail to yield reliable results in different settings due to factors like data distribution differences and unique domain-specific requirements. To overcome these limitations, it is crucial to develop more robust and flexible uncertainty estimation techniques that can adjust to the varied conditions and demands of diverse applications.

**Lack of Standardized Evaluation Protocol & Comprehensive Benchmarks for Confidence Estimation:** Current methods for evaluating uncertainty estimation methods are typically used to compare uncertainty quantification techniques through metrics like accuracy, calibration, or performance in out-of-distribution detection, using standardized datasets common within the LLM community. Despite this, variations in experimental settings across studies highlight the need for a comprehensive benchmark across tasks and domains to assess their robustness. The absence of a standardized testing protocol poses challenges for researchers from different downstream tasks, making it difficult to identify the most advanced methods or choose a specific sub-field of uncertainty quantification to pursue. This lack of uniformity hinders the direct comparison of emerging techniques and impedes the broader acceptance and integration of established uncertainty quantification methods.

## 6 CONCLUSION

In this paper, we have reviewed and analyzed the uncertainty inherent in LLMs. We clarified the definition of uncertainty and related concepts, enhancing understanding across various domains. A comprehensive framework was proposed to categorize and identify sources of uncertainty throughout the lifecycle of LLMs. We also reviewed current approaches in the literature, discussed their challenges and limitations, and highlighted future directions to enhance the practicality of LLMs in real-life applications.

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

## A  APPENDIX

You may include other additional sections here.

