# OpenReview forum: "Rethinking the Uncertainty: A Critical Review and Analysis in the Era of Large Language Models"
_ICLR.cc/2025/Conference — Submitted to ICLR 2025_

### Official Review · Reviewer_oika · 2024-10-21

**Soundness:** 3
**Presentation:** 4
**Contribution:** 2
**Rating:** 5
**Confidence:** 4

**Summary:**

This paper discusses the concept of uncertainty in the LLM era. The authors first distinguish between three often-confused concepts: uncertainty, confidence, and reliability. Then, the author delves into the uncertainty in the entire process of LLMs. Finally, the article discusses approaches for estimating and evaluating uncertainty in LLMs.

**Strengths:**

1. This article systematically discusses the concept of uncertainty in the LLM era. By clearly distinguishing between uncertainty, confidence, and reliability, the author provides a clear conceptual-level description, which is helpful for the community to have a unified discussion.

2. The authors propose a comprehensive framework for understanding uncertainty in LLMs, categorizing the uncertainties involved in LLM into operational uncertainty and output uncertainty. The article systematically and reasonably explores the uncertainty at every stages of LLM lifecycle. This well-structured classification and discussion are beneficial for understanding uncertainty.

3. The paper logic is clear. Writing is well-organized, and easy to follow. It is a very good paper to understand uncertainty in LLM era.

**Weaknesses:**

As a review and analysis-type paper, the author can enhance the technical depth and novelty by offering **deeper insights into some open questions.** In the discussion in Chapter 5, the author points out certain issues in this field. For instance, in lines 429-430, the author mentions “a more advanced framework for categorizing uncertainty estimation in large language models that surpasses the reliance on simple confidence scores.” It would be beneficial if the author could provide some insights into how such a framework could be reasonably constructed.

Another example is the issue of the “Lack of Ground Truth for Uncertainty Estimation.” I believe this is a significant challenge in the field, and it would be helpful if the author could offer some insights into potential solutions for this problem.

Generally speaking, although this paper provides a comprehensive review of LLM uncertainty, its technical depth and insights could be further enhanced to improve novelty, especially in sections 4 and 5.

**Questions:**

Please refer to the comments in the Weaknesses section.

Typo:
Line 307: importatn -> important

---

### Official Review · Reviewer_mk25 · 2024-10-24

**Soundness:** 2
**Presentation:** 2
**Contribution:** 2
**Rating:** 3
**Confidence:** 4

**Summary:**

The paper "Rethinking the Uncertainty: A Critical Review and Analysis in the Era of Large Language Models" provides a comprehensive overview of uncertainty in large language models (LLMs). It introduces a framework to categorize different types of uncertainty, which is useful for understanding and addressing challenges in critical applications of LLMs. However, the paper lacks scientific depth and novelty, failing to offer new methodologies or concrete experimental validations. The motivation for the paper is not strong enough, as it does not clearly demonstrate the practical benefits of understanding uncertainty in LLMs or how it can lead to improved performance or usability. Without empirical validation or practical examples, the theoretical framework remains abstract, which limits its value to the research community.

**Strengths:**

1. Comprehensive Overview: The paper provides a structured summary of different types of uncertainty (operational and output uncertainty) in LLMs, which helps clarify terminology and organize existing knowledge in the field.

2. Literature Review: The paper reviews various approaches for uncertainty estimation in LLMs, highlighting the strengths and weaknesses of different methods.

3. Framework for Future Work: By categorizing uncertainty, the paper lays a foundation for future research that can build upon its framework, particularly in safety-critical domains like healthcare.

**Weaknesses:**

1. Lack of Scientific Detail and Empirical Validation: The paper lacks the detailed scientific rigor required to back its claims, and there are no empirical results or experiments that demonstrate the effectiveness or utility of the proposed framework. The absence of any quantitative evaluation weakens the argument.

2. Limited Novelty: While the paper categorizes uncertainty types, it largely relies on existing concepts and frameworks, contributing little in terms of novel methodologies or groundbreaking insights. The categorization itself does not provide a fundamentally new understanding of LLM uncertainty.

3. Weak Motivation: The motivation for the paper is underdeveloped. While the authors argue that understanding uncertainty is important, they fail to convincingly explain how the proposed framework will practically improve LLM performance or utility. There is a lack of compelling use cases or scenarios that show how addressing these uncertainties will significantly benefit model reliability or interpretability.

4. No Practical Contributions: Beyond categorization, the paper does not provide practical tools, algorithms, or metrics that could be used to manage or reduce uncertainty in LLM outputs. This limits its usefulness to both researchers and practitioners.

5. Missed Opportunity in Explainability: The paper does not explore how understanding uncertainty could improve the explainability or trustworthiness of LLM outputs, which would have been a valuable contribution to areas like AI ethics or human-computer interaction.

**Questions:**

The paper falls short in delivering novel insights or practical applications that would make a significant impact in the field of LLM uncertainty estimation. The theoretical framework is not backed by empirical evidence, and the motivation lacks depth. To strengthen the paper, the authors should:

1. Provide empirical validation of the proposed framework, ideally through experiments that test how addressing uncertainty affects model performance in real-world applications.

2. Offer more concrete examples and case studies to show how understanding and managing uncertainty can lead to measurable improvements.

3. Develop and propose practical tools or algorithms that can be directly applied to LLMs for uncertainty management.

4. Strengthen the paper's motivation by showing clear benefits for LLM performance, safety, or interpretability, particularly in domains like medicine or autonomous systems.

---

### Official Review · Reviewer_govL · 2024-10-28

**Soundness:** 2
**Presentation:** 2
**Contribution:** 2
**Rating:** 3
**Confidence:** 2

**Summary:**

The paper presents a new framework for thinking about uncertainty in LLMs. To this end, it proposes standardized definitions and as well as a taxonomy for various types of uncertainty that were previously considered in the literature. The paper then presents a comparison of different existing methods for evaluating uncertainty. Based on this, future research directions are proposed in the conclusion.

**Strengths:**

- The paper considers a large body of previous work and presents it through a unified lens.
- The novel classifications and definitions are well-explained and additionally visualized with helpful figures.
- The paper will also be accessible to non-expert from the industry and policy makers.

**Weaknesses:**

# Major Weaknesses

## The paper does not demonstrate the usefulness of its proposed framework.
I am not convinced of the usefulness of the paper's claimed contributions. The paper claims its major contributions are standardized definitions and a novel framework for analyzing sources of uncertainty in LLMs. However, it is unclear to me if the framework can indeed be applied to improve research on LLM-uncertainty. In section 4, the paper evaluates different approaches for estimating uncertainty, but this section makes no use of the previously introduced framework. Instead, the comparison presented in table 1 appears to be purely a survey of existing results.

## The comparison in section 4 lacks precise metrics and seems to include subjective judgements.
Section 4 of the paper summarizes existing approaches for estimating uncertainty in LLMs into four major categories. The categories are then compared according to various metrics such as explainability, accuracy, and required compute. This comparison is summarized in a table. Unfortunately, many of the comparisons rely on vague or hard to interpret metrics. For example, what does it mean that internal-based methods can somehow be explained? What are low-, mid- or high training costs? The paper argues that these labels are relative but this is not helpful. Surely, for many of the considered metrics more precise measurements would be possible. E.g. compute costs can be operationalized as training time. Instead of accuracy, the calibration on some benchmark could be reported.

# Minor Weaknesses
- I would have like a concrete example of how "subtle nuances and variations within seemingly familiar data can still provoke uncertainties" (line 297)

# Conclusion
While the paper conducts a comprehensive survey, it seems to lack a significant contribution and at most provides a novel lens. However, it does not demonstrate the usefulness of its proposed framework and definitions. It is unclear to me how the content of section 4 and 5 benefits from the concepts introduced in section 2 and 3.  Because the paper is in essence a survey, I do not believe it provides the kind of contribution that is appropriate for acceptance to ICLR.

**Questions:**

- How, if at all, have the definitions and classifications introduced in sections 2 and 3 helped with the results and conclusions in section 4 and 5?

---

### Official Review · Reviewer_A7Ev · 2024-11-02

**Soundness:** 2
**Presentation:** 3
**Contribution:** 2
**Rating:** 3
**Confidence:** 3

**Summary:**

This paper looks uncertainties specific to LLM and designs a comprehensive framework to identify the sources and types of them. The authors demonstrate how uncertainties arise from both operational and output perspectives. For each perspective, they provide substantial prior research to support their arguments. Different approaches including internal-based and logit-based are outlined to evaluate and estimate the proposed uncertainties. They also provide future research directions on uncertainties in LLMs.

**Strengths:**

Strengths:
- The authors provide a comprehensive definition of the differences between uncertainty, confidence, and reliability, further illustrated by the plot in Figure 1. This is a crucial step to do further analysis on the uncertainties.
- The authors analyze the uncertainties not only from the training process of LLM (pre-training, instruction-finetuning, and RLHF) but also from the ambiguity in the output.
- Extensive prior research supports the framework.

**Weaknesses:**

Weaknesses:
- While I appreciate the authors are formalising the framework on uncertainties of LLMs, I find the contribution a bit weak.
- The discussion on the output uncertainties is somewhat vague. A more detailed exploration of this section and more examples would be insightful.
- Expects more numerical experiments to support the validity of the framework

**Questions:**

Questions:
- I'm a bit unclear about the definition of output uncertainties. Do these uncertainties arise from conflicts within the LLM's internal knowledge, or from the model's inability to use these knowledge in an unambiguous way? Also, what is the relationship between hallucination and output uncertainties?
- I am wondering if there is a connection between operation uncertainties and output uncertainties.

---

### Meta-Review · Area_Chair_zuqp · 2024-12-21

**Metareview:**

This paper proposed a new framework for uncertainty in LLMs. To this end, it considered standardized definitions and a taxonomy for various types of uncertainty that were previously considered in the literature. The paper then presents a comparison of different existing methods for evaluating uncertainty. The problem of interest is crucial and the authors provided a good literature review. While there are many concerns raised by reviewers, such as limited novelty, lack of scientific details, and justifications on practical utilities,  which unfortunately were not addressed at all during rebuttal. After discussion with the reviewers, we agreed it is not quite ready for publication.

**Additional Comments On Reviewer Discussion:**

All reviewers have raised similar concerns on this paper regarding the limited novelty, lack of scientific details, and justifications on practical utilities. The authors have not responded to this criticism and we agreed it is not quite ready for publication.

---

### Decision · Program_Chairs · 2025-01-22

Reject